# Coherent Stokes Raman scattering microscopy (CSRS)

**Sandro Heuke** [1] ✉ **& Hervé Rigneault** [1] ✉

We report the first implementation of laser scanning coherent Stokes Raman scattering (CSRS) microscopy. To overcome the major challenge in CSRS imaging, we show how to suppress the fluorescence background by narrow bandpass filter and a lock-in based demodulation. Near background free CSRS imaging of polymer beads, human skin, onion cells, avocado flesh and the wing disc of a drosphila larva are presented. Finally, we explain and demonstrate numerically that CSRS solves a major obstacle of other coherent Raman techniques by sending a significant part (up to 100%) of the CSRS photons into the backward direction under tight focusing conditions. We believe that this discovery will pave the way for numerous technological advances, e.g., in epi-detected coherent Raman multi-focus imaging, real-time laser scanning based spectroscopy or efficient endoscopy.

Conventional bright-field microscopy provides information about the refractive index and absorption properties but cannot elucidate the sample's chemical composition. Infrared absorption and linear Raman scattering can provide the sample chemical composition[1,2], but they are incompatible with high spatial resolution or real-time imaging. Coherent Raman scattering imaging (CRS) fills this gap by combining chemical sensitivity with signal levels that permit video-rate image acquisition. Well-established CRS microscopy techniques are coherent anti-Stokes Raman scattering (CARS)[3,4] and stimulated Raman scattering (SRS)[5–7]. CARS owe its wide-range application to the blue-shifted anti-Stokes radiation, which greatly facilitates its separation from linear fluorescence. When working with near-infrared excitation wavelengths, the blue-shifted CARS radiation is readily detected using photo-electron multiplier tubes (PMT) of standard laser scanning microscopes. SRS's popularity arises from the heterodyne signal amplification that frees SRS images from an omnipresent non-resonant four-wave-mixing background that is present in CARS images[8]. SRS also allows for measurements under daylight conditions owing to its modulation and signal detection scheme.

Overshadowed by CARS and SRS until now, there exists a third four-wave-mixing process that can be resonant with vibrational levels termed coherent Stokes Raman scattering (CSRS, pronounced "SCiSsoRS")[8–12]. CSRS, as CARS and SRS, is always present in CRS experiments and also provides chemical sensitivity[13]—see Fig. 1. In analogy to the Stokes emission in linear Raman microscopy, the CSRS radiation $(2\omega_S - \omega_P)$ is red-shifted with respect to the excitation

frequencies of the pump ($\omega_P$) and Stokes beams ($\omega_S$). Surprisingly, CSRS imaging has never been implemented for laser scanning microscopy (LSM). Presumably, this is due to the high degree of resemblance of CARS and CSRS spectra[13], rendering CSRS—prima facie—to be either CARS with an added fluorescence background when working with visible light sources or CARS with a radiation wavelength offside high quantum yields of common detectors when working with near-infrared (NIR) excitation. CSRS provides, however, some unique properties that are of high interest for imaging.

First, the CSRS spectrum differs from CARS in the presence of accessible electronic resonances. For example, pre-resonant CSRS will offer complementary information in the application of alkyne-labeled dyes[14] and standard dyes used in microbiology[15]. Second, the red-shifted radiation of CSRS becomes an advantage for UV or near-UV excitation where CARS photons[16] would be too far blue-shifted to be detected efficiently while any SRS image[17] is likely to be compromised by various artifacts such as multi-photon absorption[18,19]. Thus, UV-excited CSRS holds the potential to achieve the highest possible spatial resolution ($\lambda_{Stokes}/[\sqrt{8}NA]$) in coherent Raman imaging. Third, NIR-excitation wavelength combined with CSRS may allow for deeper tissue imaging due to the reduced scattering and absorption of its radiation[20]. Last but most important: Due to a modified phase-matching geometry, CSRS microscopy can be configured to radiate more light in the backward direction. This game changer would benefit the investigation of thick samples, real-time spectroscopy, multi-focus imaging, and endoscopy[21]. Within this contribution, we want to open

[1]Aix Marseille Univ, CNRS, Centrale Marseille, Institut Fresnel, Marseille, France. ✉e-mail: sandro.heuke@fresnel.fr; herve.rigneault@fresnel.fr

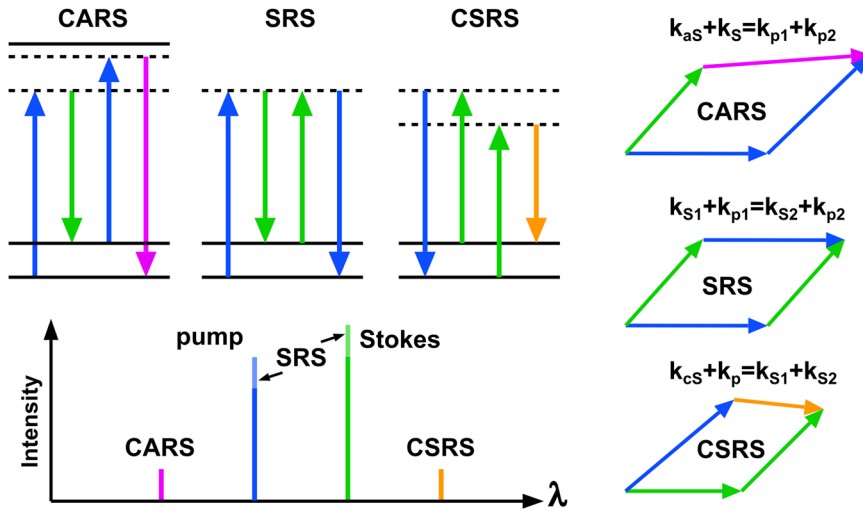

**Fig. 1 | Overview coherent Raman imaging techniques.** In energy diagrams[33], relative radiation wavelength and energy conservation under plane-wave illumination.

up the field of laser scanning CSRS imaging by demonstrating CSRS microscopy within the visible excitation spectrum. To remove the major fluorescence background obstacle, we will show how linear fluorescence can be suppressed by combining a set of bandpass filters with a lock-in-based detection scheme. Furthermore, we shall investigate numerically CSRS' spatial radiation behavior under NIR excitation, paving the way towards CSRS experiments with an efficient epi-detection.

## Result and discussion

### Experiments

The CSRS signal of biomedical samples is often overwhelmed by linear fluorescence as CSRS radiation is red-shifted as compared to the excitation lasers. Time-gating[22], time-resolved detection using streak cameras[23], or polarization filtering can be used to reduce or suppress any fluorescence signal. However, these methods require either a substantial modification of standard coherent Raman microscopes or do not work in the presence of strong fluorescence light backgrounds. Here, we exploit the fact that the CSRS is spectrally narrow under ps-excitation. Thus, the majority of fluorescence can be readily suppressed by narrow-band filters. Filters with a spectral width below <1 nm are commercially available, but the selection of a specific center wavelength requires expensive custom solutions. Instead, we use a combination of two inexpensive bandpass filters with a spectral width of about 15 nm but with different center wavelengths. In addition, we fine-tune the filter transmissions by tilting them (<20°) with respect to the incident beam, see Fig. 2b. Thus, two tilt-adjusted bandpass filters create a sharp transmission line (FWHM < 3 nm) for the CSRS signal collection while rejecting a significant part of the autofluorescence.

As a second method for fluorescence background rejection, we take advantage of the CSRS intensity dependence on both pump and Stokes excitation colors while linear fluorescence follows either the intensity of the pump or the Stokes laser, see Fig. 2c. Consequently, modulating the pump and Stokes beams at f1 and f2 while demodulation the signal at f1−f2 (or f1 + f2) yields exclusively nonlinear signals that depend on both excitation colors. The f1−f2 demodulation, therefore, also discriminates the CSRS signal against 2-photon excited fluorescence (2PEF) under single-color excitation. It should be noted that the f1−f2 modulation is also sensitive to two-color 2-photon fluorescence (2C-2PEF). Nevertheless, we will find experimentally that the emission strength of native 2C-2PEF is negligible within our CSRS implementation using visible beams.

For the experimental implementation of CSRS LSM, we chose visible excitation wavelengths at 445 nm (pump) and 515 nm (Stokes)

for the following reasons: First, CSRS under near UV excitation is a potentially important application area since CARS, and SRS encounter experimental difficulties within this spectral range: the CARS signal falls into the UV range while SRS artifacts are increased due the possible high concentration of endogenous chromophores. Second, the red-shifted CSRS radiation can be readily detected by ordinary PMTs. Third, stress test: fluorescence artifacts are enhanced as compared to a near-infrared (NIR) excitation. Thus, our approach will be viable as well for CSRS under NIR excitation if pure CSRS signals can be obtained under VIS excitation.

The experimental setup, the spectral filtering, and the double modulation are schematically shown in Fig. 2a. Our implementation resembles a standard SRS setup with the difference that we use visible excitation wavelengths, we modulate not one but both beams, and the photo-diode is replaced by a PMT which is connected to a lock-in amplifier. More information about the setup can be found in part "Methods: Experimental setup". To quantify the level of fluorescence rejection, we investigated the signal of native olive oil at 2850 cm⁻¹ when blocking the pump or Stokes beams or when the temporal pulse overlap is removed. The output signal of the lock-in is plotted as functions of the demodulation frequencies at 0 Hz (DC), f1, f2 and f1−f2 in Fig. 2d. It can be observed that the DC channel contains significant amounts of fluorescence while this artifact is already reduced within the f1 and f2 channels. Nevertheless, only the difference frequency channel at f1−f2 approaches zero when the excitation pulses do not overlap in time (Δt ≫ 3ps). In a second experiment, we imaged with CSRS (demodulated at f1−f2) the interface between olive oil and a 20 μm sized Plexiglas (PMMA) bead to obtain an estimation of the lateral resolution with an excitation objective of NA = 1.45− see Fig. 2e. From this "knife-edge" CSRS intensity profile, we can infer a lateral resolution below 400 nm. The difference to the expected $\lambda_{\text{Stokes}}/[\sqrt{8}\text{NA}] = 515$ nm/ $[\sqrt{8}1.49] = 120$ nm can be attributed to the underfilling of the excitation objective back aperture and the bent oil/bead interface. Having confirmed a high-resolved, fluorescence-free CSRS image contrast, we investigated the suitability of LSM-CSRS for vibrational imaging of various objects featuring non-negligible background fluorescence levels. Within Fig. 3, we show the CSRS images of test and biomedical samples demodulated at the DC and f1-f2 frequencies for (non-)overlapping pump and Stokes pulses. The images were organized along the ratio of the CSRS to the fluorescence signal, starting from the highest at the top. Comparing the DC and f1−f2 images in Fig. 3a, it is obvious that narrow spectral filtering is already sufficient for CSRS imaging of polymer beads in oil (see

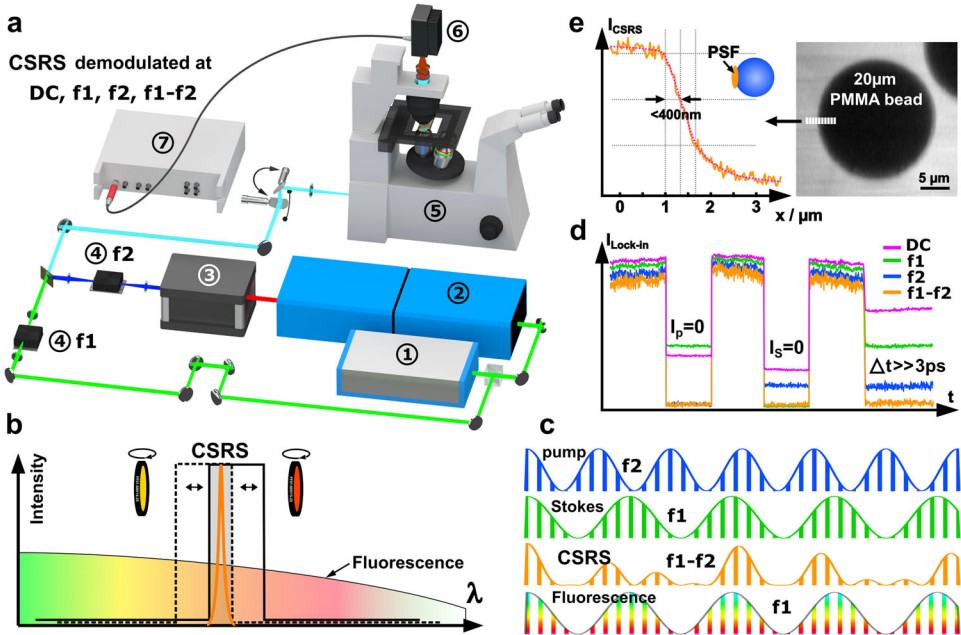

**Fig. 2 | CSRS experimental implementation and characterization. a** Scheme of the CSRS experiment. 1. Yb-fiber laser, 2. optical parametric oscillator (OPO), 3. second harmonic generation (SHG), 4. acousto-optic modulator (AOM), laser scanning microscope (LSM), 6. photo-electron multiplier (PMT), 7. lock-in amplifier. **b** The CSRS signal is separated from fluorescence by means of two angle-tuned narrow bandpass filters. **c** Additional suppression of fluorescence is achieved by intensity modulating the Stokes and pump beam at the frequencies f1 and f2,

respectively. Fluorescence-free CSRS signal is obtained at f1−f2. **d** Measured CSRS signal at DC, f1, f2, and f1−f2 frequencies when the pump or Stokes beam is blocked (Ip = 0 or Is = 0) or when their temporal overlap is removed (Δt ≫ 3 ps). The fluorescence is strongly rejected on the f1−f2 time trace and mainly comes from the Stokes beam. **e** The CSRS intensity profile obtained at f1−f2 at the interface of a PMMA bead and olive oil indicates a lateral resolution of <400 nm.

CSRS at DC). The first artifacts become visible for the DC CSRS images of the epithelium and dermis of a 20 μm thick section of human skin—see Fig. 3b, c. For the epithelium, a pronounced fluorescence artifact arises from melanin within the epidermis dermis junction. Artifacts within the dermis can be attributed to the autofluorescence of collagen and elastin[24]. The quantity of fluorescence observed within the DC channel increases stepwise further for CSRS imaging of onion cells, lipid droplets within the flesh of an avocado, and the wing disc of a Drosophila larva. From the second row of Fig. 3, we validate that almost no fluorescence is leaking into the f1−f2 CSRS channel.

In the next section, we address a non-intuitive but key feature of CSRS microscopy: the possibility to dramatically increase the CSRS backwards radiation opening the road for an effective epi-CSRS detection.

## Momentum conservation and simulations

Before entering into the calculations, we want to consider CSRS from a heuristic viewpoint investigating the momentum conservation laws for CSRS and compare it to CARS. Under plane illumination, the momentum conservation laws can be written as $\mathbf{K} = \mathbf{k}_p - \mathbf{k}_S + \mathbf{k}_p - \mathbf{k}_{aS}$ for CARS[25] and $\mathbf{K} = \mathbf{k}_S - \mathbf{k}_p + \mathbf{k}_S - \mathbf{k}_{cS}$ for CSRS with $\mathbf{K}, \mathbf{k}_p, \mathbf{k}_S, \mathbf{k}_{aS}$ and $\mathbf{k}_{cS}$ representing the wavevectors of the object, the pump (probe) and Stokes beam as well as the anti-Stokes and coherent Stokes radiation, respectively. Note that for homogeneous samples ($\mathbf{K} = 0$) these laws are also referred to as phase-matching condition and simplify to $\mathbf{k}_p + \mathbf{k}_p = \mathbf{k}_S + \mathbf{k}_{aS}$ (CARS) and $\mathbf{k}_S + \mathbf{k}_S = \mathbf{k}_p + \mathbf{k}_{cS}$ (CSRS). Under focusing conditions, the single wavevectors are replaced by the distribution of incident wavevectors which are distributed over a cap of a sphere . To identify those object frequencies ($\mathbf{K}$) that are effectively probed, every combination of excitation and emission wavevector must be identified. This operation is equivalent to the convolution of the caps of the illumination and detection Ewald spheres[26]. Neglecting polarization

effects, the result of this convolution (simplified to 3 points per arc) is shown in 2D within Fig. 4a.

Evidently, there exists no vector combination for epi-scattered CARS photons which covers the origin $K(0,0,0)$ of the object space. Thus, a homogeneous sample, such as olive oil, does not provide any backward CARS radiation. On the contrary, structures that feature high object frequencies, such as small polymer beads or layered materials, generate Epi-CARS radiation. In the past, Epi-CARS was occasionally considered to be a size-selective contrast that would highlight exclusively small objects[27]. While this statement holds for the majority of biomedical samples, there do exist large structures, e.g., multi-layered lipids in vesicles, that also emit a strong CARS radiation in the backward direction. Hence, it is more appropriate to refer to Epi-CARS as a technique that probes high object frequencies along the z-axis instead of being considered as size-selective.

Switching the detection wavelength to the red-shifted CSRS radiation changes the covered object support significantly and includes now the origin at K(0,0,0). Due to the reduced size of the detection wavevector ($|\mathbf{k}_{cS}| \ll |\mathbf{k}_{aS}|$), steep incident angle Stokes vectors, and the pump vector entering as complex conjugated, it is now possible to find vector combinations that cover the origin at $K(0,0,0)$. Consequently, even a homogeneous object will radiate considerable amounts of Epi-CSRS. Nevertheless, since the centroid of the Epi-CSRS object support, i.e., the gray cloud within Fig. 4a, does not coincide with the K-space origin $K(0,0,0)$, Epi-CSRS images will also highlight objects containing higher frequencies.

To address the question of how to increase the ratio of backward versus forward CSRS and which object frequencies are most efficiently probed using Epi-CSRS, we performed finite element simulations whose results are summarized in Fig. 4b-e. The equations implemented numerically, as well as parameters, are found in methods: numerical calculation. From the momentum conservation law and the vector diagrams in Fig. 4a, it is readily comprehensible that a larger wavelength difference between the pump and CSRS wavelength greatly

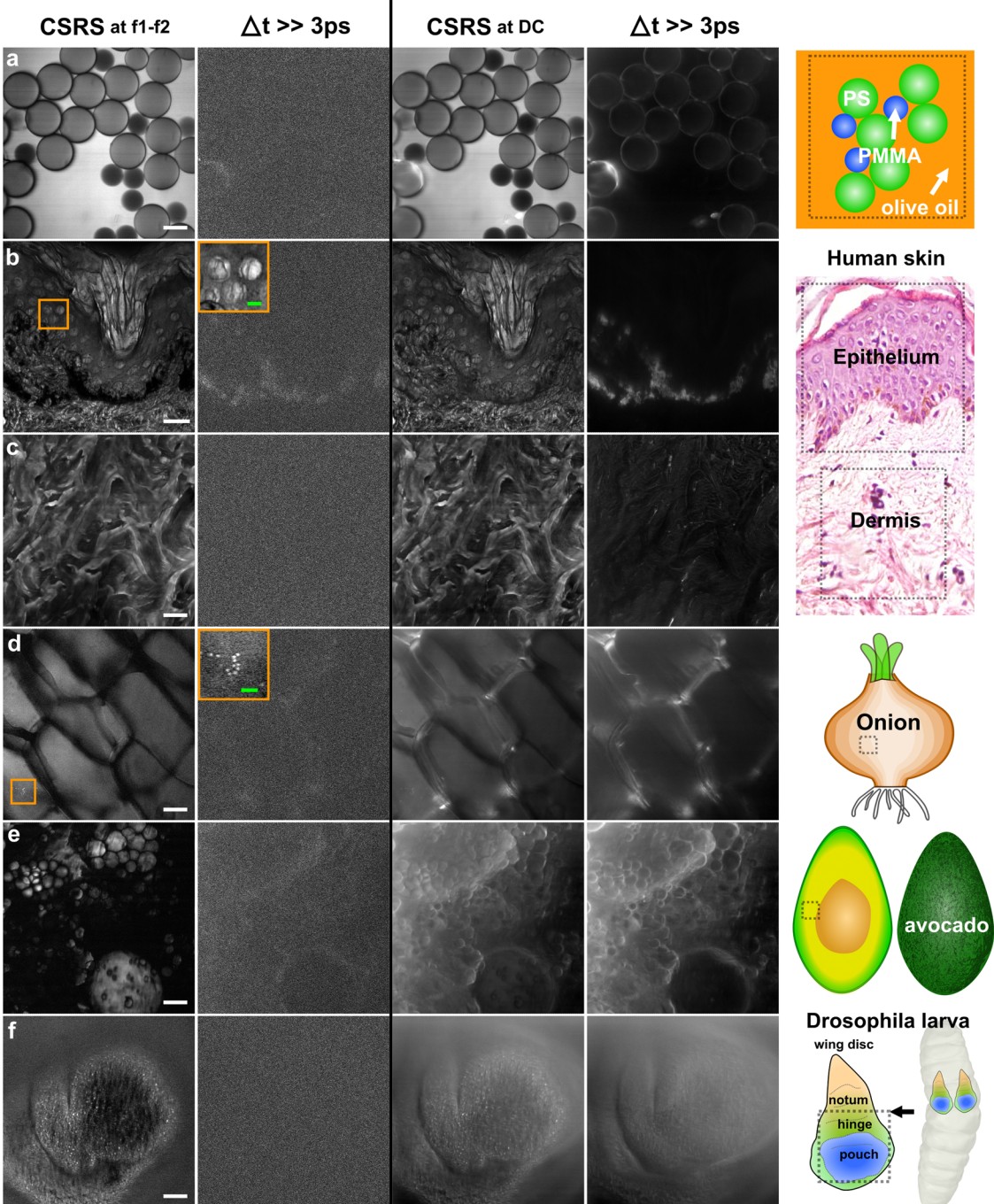

**Fig. 3 | Laser scanning CSRS at 2850 cm⁻¹.** The left and right column show the CSRS image demodulated at the frequencies f1–f2 = 1.47 MHz and 0 Hz (DC), respectively. To estimate the remaining fluorescence level, images without temporal overlap of the pump and Stokes pulses are displayed to the right (Δ*t* ≫ 3ps). **a** Mixture of polystyrene (PS, 30 μm) and Poly-methyl-methacrylate (PMMA, 20 μm) beads in olive oil. **b, c** Epithelium and dermis of a 20 μm thick human skin section. **d** Cells of an onion. **e** Lipid droplets within the flesh of an avocado. **f** Wing disc of a *Drosophila* larva. The insets displayed in the second column are the zoomed "CSRS at f1–f2" regions of interest shown in the first column on (**b, d**). Pixel dwell time: 40 μs. Image acquisition time: 40 s (1000 × 1000). The white and green scale bar equals 20 and 5 μm, respectively.

relaxes the necessity for extreme incident illumination angles of the Stokes beam. The wavelength difference between the pump and CSRS radiation is enhanced using NIR instead of VIS excitation wavelengths, which is why we used in our simulations the wavelength $\lambda_p = 797$ nm and $\lambda_S = 1030$ nm, which matches the 2850 cm⁻¹ Raman shift. For these conditions, the coherent Stokes radiation is observed at $\lambda_{cS} = 1450$ nm. It should be noted that our results equally apply to the visible excitation wavelength using a higher excitation angle (or thinner annular masks—see below).

To start with, we computed the radiation pattern of CSRS and CARS of a homogeneous object using an NA of 1.49 (oil immersion), corresponding to a maximum illumination angle of 80°. From Fig. 4b, it is evident that both CARS and CSRS are predominantly forward directed though the CSRS' radiation distribution features a larger radiation cone. Considering the ratio of backward versus forward-directed photons $R_{b/f}$, we find numerically that less than 1 photon in 10⁵ is backward-directed for CARS. Note that the momentum conservation law actually predicts $R_{b/f} = 0$ for CARS. Thus, the deviation observed

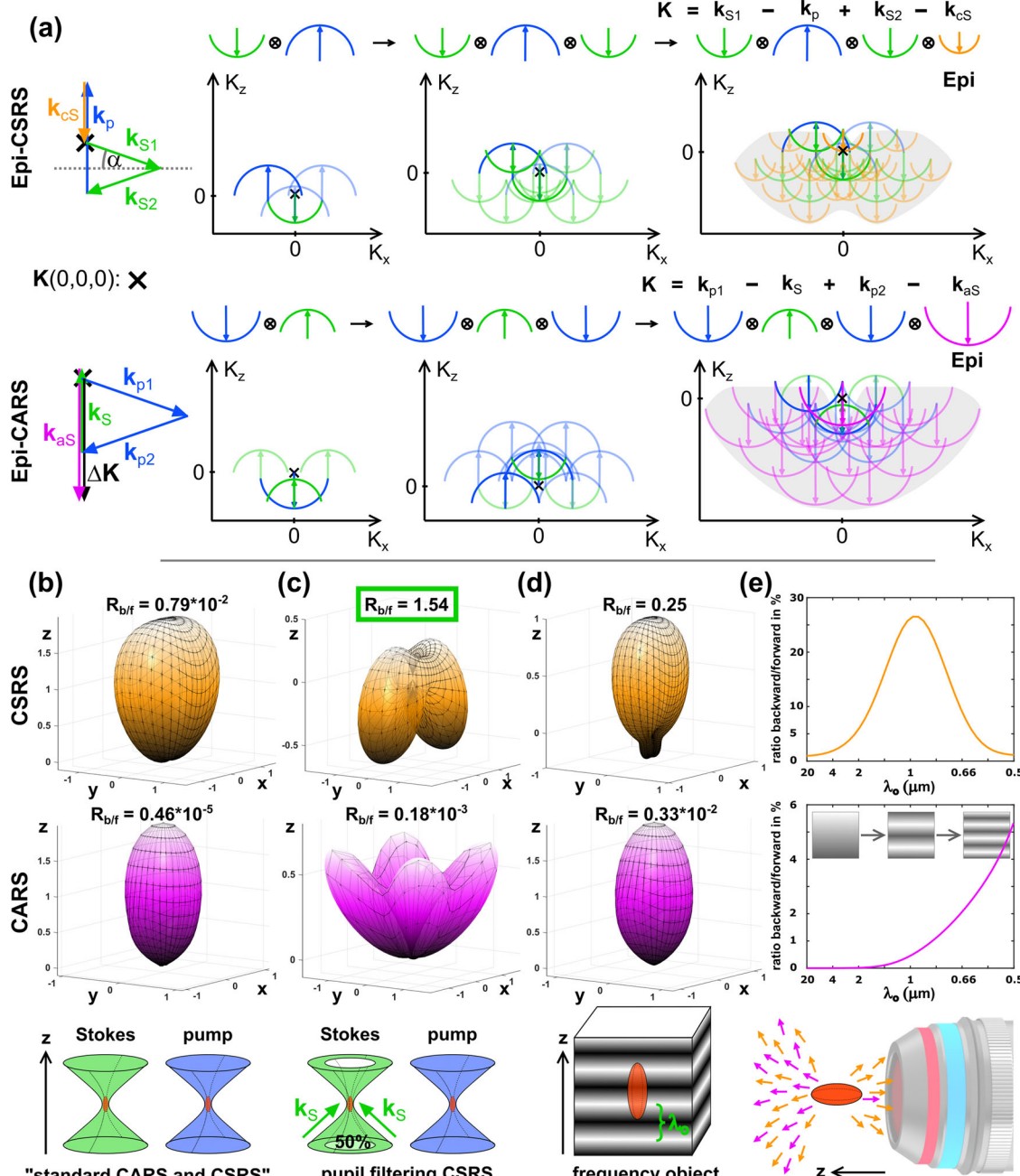

**Fig. 4 | Object frequency support and radiation behavior of CSRS versus CARS.**
**a** The object spatial frequency K-support for Epi-CSRS(CARS) is found by convolving the illumination Ewald spheres of the Stokes (pump), pump (Stokes), and Stokes (probe) with the cap of detection Ewald sphere at coherent Stokes (anti-Stokes) frequency. Note that vector combinations covering the frequency of a homogeneous sample $K(0,0,0)$ are only found for CSRS but not for CARS. A single wavevector combination that phase-matches $K(0,0,0)$ is highlighted to the left, while a similar approach for CARS leads to a large phase-mismatch ($\Delta K$). **b** CSRS and CARS radiation behavior of a homogeneous sample under standard illumination

conditions, i.e., the pump and Stokes beam fill the objective aperture homogeneously ($\theta_{max} = 80°$). **c** same as in **b** but with an annular pupil filter applied to the Stokes beam for CSRS covering 50% of the area of the objective back-aperture. For an equitable comparison with CARS, the same pupil filter was applied to the pump beam. **d** same as for **b** (conventionally focused beams), but the homogeneous sample was replaced by a frequency object whose scatter density is described as $1 + \cos(2\pi z/\lambda_o)$ and $\lambda_o = 1\,\mu m$. **e** Plot of the ratio of backward/forward radiation ($R_{b/f}$) as a function of the object frequency $\lambda_o$. Calculations were performed with $\lambda_p = 797\,nm$ and $\lambda_S = 1030\,nm$.

must be attributed to the finite number of voxels of the numerical model. For CSRS, $R_{b/f}$ increases dramatically to about 1 in 100 photons. Since common surfaces within biomedical samples scatter more than 1%, it is still likely that in this high NA illumination scheme, epi-detected CSRS is dominated by forward-generated CSRS that is back-scattered by linear scattering at interfaces (as in the CARS case). To find an approach that increases the proportion of epi-CSRS radiation, we consider the CSRS vector diagram matching $K(0,0,0)$ on the top left of

Fig. 4a. The ratio of backward versus forward radiation is readily increased by reducing the impact of vectors combinations probing higher frequencies and favoring those that cover the origin by satisfying $\mathbf{k}_S + \mathbf{k}_S = \mathbf{k}_p + \mathbf{k}_{cS}$. This enhancement of epi-CSRS radiation can be achieved using an annular illumination of the Stokes beam. Experimentally, such an annular illumination can be generated, without power loss, using two axicons within the Stokes beam path[28,29]. Numerically, we restricted the incident angles for the Stokes between

$\theta_{min} = 56.5°$ and $\theta_{max} = 80°$, which corresponds to covering 50% of the area of the objective lens' back-focal plane. The pump beam remains a normally focused beam and covers the full lens' back-focal plane. With this Stokes pupil filtering, the ratio of backward to forward radiation increased for CARS to 2 in $10^4$ photons while most of the CSRS radiation is backward directed ($R_{b/f} = 1.5$) when the object is homogeneous—see Fig. 4c. For the CARS calculation we considered the annular illumination applied to the pump beam whereas the Stokes is a conventional focused beam.

As a second important result from the heuristic derivation of CSRS object support, we found that the presence of high spatial frequencies along $K_z$ increases the amount of backward radiation. To confirm this prediction, we investigated in Fig. 4d, e an object whose nonlinear scatterer density, i.e., the concentration of molecular groups, is modulated along the optical axis as $1 + \cos(K_z z)$ with $K_z = 2\pi/\lambda_o$ being the object frequency. We now consider a conventional illumination scheme where both Stokes and the pump are tightly focused and cover the full back aperture of the objective lens. Figure 4d outlines the radiation behavior of such a z-structured object with $K_z = 2\pi/1\,\mu m$. It is found that $R_{b/f}$ increases to one-fourth for Epi-CSRS while Epi-CARS remains negligible weak. To identify those object frequencies which are most efficiently probed by Epi-CSRS, we computed $R_{b/f}$ as a function of $K_z$. From Fig. 4e, we find that Epi-CSRS peaks at $K_z = 2\pi/1\,\mu m$ whereas Epi-CARS $R_{b/f}$ still increases at $K_z = 2\pi/0.5\,\mu m$ confirming that CARS requires larger $K_z$, i.e., objects with higher frequency modulation of the scatterer density, to generate a strong Epi radiation.

Thus, we have found that CSRS features non-negligible backward radiation from a homogenous sample under tight-focusing conditions, while this is not the case for CARS. The amount of backward radiated CSRS can be further enhanced using a Stokes annular illumination to surpass the forward CSRS radiation.

In conclusion, we have demonstrated the first LSM CSRS experiment. As the major challenge, we were able to reduce the fluorescence background significantly using a pair of tilted bandpass filters. The remaining fluorescence contribution was removed by intensity modulating the Stokes and pump beams at the frequencies f1 and f2 and a lock-in-based demodulation of the CSRS signal. Taking advantage of CSRS' characteristic dependence on both excitation colors, near fluorescence-free CSRS images were obtained when demodulating the CSRS signal at f1–f2. Fluorescence-free LSM-CSRS imaging was demonstrated on a variety of samples showing different fluorescence levels, such as polymer beads, epithelium, and dermis of human skin, onion cells, avocado flesh, and the wing disc of *Drosophila* larva. Having demonstrated the viability of CSRS imaging, we introduced and quantified numerically how CSRS can be implemented to generate a strong backward radiated signal with high NA objective lenses. CSRS' unique backward radiation ability can be understood considering the momentum conservation laws for all combinations of all contributing k-vectors and cannot be achieved with CARS or SRS. With efficient backward radiation at hand, various coherent Raman experiments become feasible, which were impossible before. For example, this is the case for Epi-detected confocal multi-focus CSRS, Epi-detected LSM-CSRS with a spectrometer at the descanned position, Epi-detected CSRS image scanning microscopy, or efficient endoscopy. Thus, we believe that this discovery will open new directions for coherent Raman developments and applications.

## Methods
### Experimental setup
A Yb-based fiber laser (APE Emerald engine, 80 MHz, 2–3 ps) is frequency doubled, yielding 7 W of 515 nm output power. Parts of the emissions are used directly as a Stokes beam to drive the CSRS process. The major part (4 W) of the 515 nm is employed to pump an

optical parametric oscillator (OPO, APE Emerald). The OPO's signal beam is tunable to 660–950 nm and coupled into an external SHG unit (APE, HarmoniXX). The latter generates up to 50 mW within the spectral range of 330–475 nm and serves as the pump beam for CSRS. Thus, the 330–475 nm pump combined with the 515 nm Stokes beam allows addressing a Raman shift range from 1630–11,000 cm⁻¹. The pump and Stokes beams are superimposed in space and time using a dichroic beam splitter (Semrock, FF470-Di01-25x36) and a delay stage. Both beams are coupled into a home-built laser scanning microscope and focused by a 40× water objective lens (Nikon, Plan, NA = 1.15, immersion: water) into the sample. The excitation objective lens was replaced for a 60× objective (Nikon, Plan Apo TIRF, NA 1.45, immersion:oil) to generate the bead-oil interface image within Fig. 2. The CSRS radiation is collected by a condenser lens (Nikon, Achr-Apl, NA 1.4) in the forward direction, spectrally separated from the broadband fluorescence background by means of 2 tilted bandpass filter (Semrock FF01-620/14–25 + FF01-605/15–25) and detected by a photo-electron multiplier (PMT, Thorlabs, PMT1001). We measured the CSRS and CARS (at 398 nm) radiation strength for olive oil one after another and found comparable signal levels. To avoid detector saturation for the acquisition of CSRS images, we applied the lowest possible PMT gain corresponding to an amplification of only $5 \times 10^3$. For an enhanced suppression of the linear fluorescence background, two acousto-optic modulators (AOM, AA, MT200-A0.5-VIS) were applied to modulate the intensity of the Stokes and pump beams and at the frequencies f1 = 2.28 MHz and f2 = 3.75 MHz, respectively. The PMT output was demodulated simultaneously at the DC frequency, f1, f2, and at f1–f2 = 1.47 MHz using a lock-in amplifier (Zürich instruments, HF2LI). The lock-in time constant was set to 30 µs. All CSRS images shown were recorded with a pixel dwell time of 40 µs. All samples were investigated in live image acquisition mode. Thus, some areas were scanned more than 100 times. We noticed that the fluorescence background signal within the DC channel was reduced over time as a result of photo-bleaching though the f1–f2 channel remained unaffected, which indicates that our experimental conditions are below the damage threshold of ex vivo samples. Note that the demodulation at f1–f2 only removes the fluorescence background while the CSRS non-resonant four-wave-mixing background[30] that is inherent to all coherent Raman techniques is still present. Nevertheless, the removal of this non-resonant background could be achieved by a heterodyne interference of the CSRS signal with a reference beam at the same wavelength[31] or by Kramers–Kronig or Maximum entropy-based algorithms in application to CSRS spectra[12].

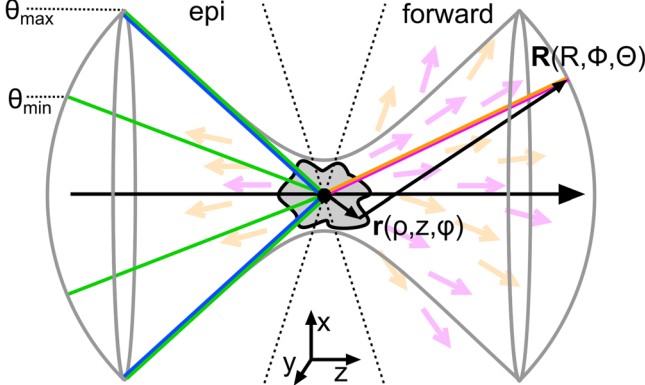

**Fig. 5** | Declaration of variables.

## Numerical calculation

In the following, we shall summarize the equations used to generate Fig. 4b–e. The meaning of the variables is summarized in Fig. 5.

The focused field at the sample is given by the angular spectrum representation[32]:

$$\begin{bmatrix} E_x(\rho,\phi,z) \\ E_y(\rho,\phi,z) \\ E_z(\rho,\phi,z) \end{bmatrix} = \frac{ikf}{2}\exp(-ikf)\begin{bmatrix} I_{00}+I_{02}\cos(2\phi) \\ I_{02}\sin(2\phi) \\ -i2I_{01}\cos(\phi) \end{bmatrix} \quad (1)$$

Here $f$ denotes the focal length of the objective lens, and the integrals $I_{0m}$ are provided by

$$I_{0m} = \int_{\theta_{\min}}^{\theta_{\max}} E_{\text{inc}}(\theta)\sin(\theta)[\cos(\theta)]^{1/2}g_m(\theta)J_m[k\rho\sin(\theta)]\mathrm{d}\theta \quad (2)$$

where $g_m$ equals $1+\cos(\theta),\sin(\theta)$ and $1-\cos(\theta)$ for $m = 0, 1, 2$, respectively. $J_m$ is the $m$th order Bessel function while $E_{\text{inc}}$ is the incoming electric field which we assumed to be $x$-polarized and constant within the (annular) aperture angles $\theta_{\min} \leq \theta \leq \theta_{\max}$. The nonlinear polarization at anti-Stokes and coherent Stokes wavelength is given by:

$$\begin{aligned} P_{aS,a}^{(3)}(r) &= 3\chi_{abcd}^{(3)}(r)E_{p,b}E_{S,c}^*E_{p,d} \\ P_{cS,a}^{(3)}(r) &= 3\chi_{abcd}^{(3)}(r)E_{S,b}E_{p,c}^*E_{S,d} \end{aligned} \quad (3)$$

Where a,b,c,d represent the polarization coordinates $x$, $y$, or $z$. Using an $x$-polarized excitation, it was noticed that $\chi_{xxxx}^{(3)}$ dominates all other tensor components even under tight focusing conditions while filling the objective lens homogeneously[32]. Nevertheless, for the generation of Fig. 4c, an annular mask with $\theta_{\min} = 56.5°$ and $\theta_{\max} = 80°$ was applied, which does necessitate the inclusion of other tensor elements. For simplicity, we consider here only isotropic samples reducing the 81 susceptibility tensor elements to 21, which are nonzero[30]. Within isotropic media, these nonzero elements follow certain symmetry rules which are, $\chi_{1111} = \chi_{2222} = \chi_{3333}$, $\chi_{1122} = \chi_{1133} = \chi_{2211} = \chi_{2233} = \chi_{3311} = \chi_{3322}$, $\chi_{1212} = \chi_{1313} = \chi_{2323} = \chi_{2121} = \chi_{3131} = \chi_{3232}$, $\chi_{1221} = \chi_{1331} = \chi_{2112} = \chi_{2332} = \chi_{3113} = \chi_{3223}$. Further, it applies $\chi_{1111} = \chi_{1122} + \chi_{1212} + \chi_{1221}$[30]. Within our simulations we were setting $\chi_{1122} = \chi_{1212} = \chi_{1221} = 1$ and, hence, $\chi_{1111} = 3$. The nonlinear far-field radiation distributions are obtained using a dyadic Green function approach:

$$\begin{aligned} &\begin{bmatrix} E_{q,R}(R,\Theta,\Phi) \\ E_{q,\Theta}(R,\Theta,\Phi) \\ E_{q,\Phi}(R,\Theta,\Phi) \end{bmatrix} = -\frac{\omega_q^2}{c^2}\frac{\exp(ik_q|R|)}{|R|}\iint\int_{-\infty}^{\infty}\rho\mathrm{d}\rho\mathrm{d}\phi\mathrm{d}z\frac{\exp(ik_q\mathbf{rR})}{|R|} \\ &\times \begin{bmatrix} 0 & 0 & 0 \\ \cos(\Theta)\cos(\Phi) & \cos(\Theta)\sin(\Phi) & -\sin(\Theta) \\ -\sin(\Phi) & \cos(\Phi) & 0 \end{bmatrix}\begin{bmatrix} P_{q,x}^{(3)}(\mathbf{r}) \\ P_{q,y}^{(3)}(\mathbf{r}) \\ P_{q,z}^{(3)}(\mathbf{r}) \end{bmatrix} \end{aligned} \quad (4)$$

where $q$ is replaced by aS or cS to calculate either the anti-Stokes or coherent Stokes radiation. Within the simulations, we segmented the focal area into ($121 \times 121 \times 121 \approx$) 1.77 million elements of a width of 50 nm equally spaced into the $x$, $y$, and $z$ directions. The far-field radiation sphere was discretized into ($\Delta\Theta = 1°$, $\Delta\Phi = 2°$) 32,400 elements. The coherent (anti-)Stokes radiation was qualified as either forward or backward-directed if falling into the range $\Theta.. 0$–$80°$ or $\Theta.. 100$–$180°$, respectively.

## Data availability

The data that support the findings of this study are available from the corresponding author upon request.

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

## Acknowledgements

We acknowledge financial support from the Center National de la Recherche Scientifique (CNRS, S.H., H.R.), A*Midex (ANR-11-IDEX-0001-02, H.R.), ANR grants (ANR-10-INSB-04-01, ANR-11-INSB-0006, ANR-16-CONV-0001, ANR-21-ESRS-0002 IDEC, H.R.), INSERM PC201508, 18CP128-00, 22CP139-00, (H.R.). This project has received funding from European Union's Horizon 2020 EU ICT 101016923 CRIMSON (H.R.).

## Author contributions

S.H. conceived the idea, and performed the experiments and numerical calculations. H.R. conceived the idea and discussed the results. S.H. and H.R. wrote the paper.

## Competing interests

The authors declare no competing interests.
