## [Peer Review File · Nature Communications]

Coherent Stokes Raman scattering microscopy (CSRS)Reviewer #1 (Remarks to the Author):

Heuck et al. present an implementation of CSRS where the pump and Stokes driving fields are independently modulated at f_1 and f_2 , and the coherent Stokes signal field is detected through a spectral notch filter at $f_1 - f_2$. They also perform calculations of the angular dependence of the signal field under standard high numerical aperture focusing and with pupil-filtered illumination.

The description of the experiment is clear, and the modulation scheme is similar to some of their previous work. Figure 3 shows that their fluorescence filtering approach is effective, and their signal field calculations suggest an interesting effect in an enhanced epi-emitted signal.

I have a few comments:

1) While this is a very nice example of CSRS, it is not the first time that CSRS imaging has been demonstrated. Bito et al showed in 2013 that simultaneous CARS and CSRS imaging was beneficial for interpreting spectroscopic signals. That work should be cited: Bito et al., "Three-pulse multiplex coherent anti-Stokes/Stokes Raman scattering (CARS/CSRS) microspectroscopy using a white-light laser source, Chemical Physics, Volume 419, 2013.

2) The diagram for CSRS shown in Figure 1 is different than I have seen elsewhere (such as in their reference 11 and in the Bito paper cited above). If this is a non-standard representation, perhaps they could comment on that.

3) The authors should indicate the time required to acquire the images. This information could be placed in the caption of Figure 2.

4) While the calculation results shown in Fig. 4 are interesting, it would be appropriate for the authors to demonstrate the effect they predict since it is the most unique aspect of the work.

Reviewer #2 (Remarks to the Author):

This paper provides the first experimental demonstration of a novel coherent Raman imaging modality, Coherent Stokes Raman Scattering (CSRS). In CSRS a light red-shifted with respect to the Stokes frequency ω_S is generated, according to the process $\omega_{CSRS} = 2\omega_S - \omega_{pu}$. Detection of the CSRS signal is challenged by its overlap with the sample fluorescence, generated by the pump/Stokes beams. Here the authors suppress the fluorescence by a combination of narrow bandpass filtering and modulation followed by lock-in detection, and demonstrate CSRS imaging on a series of test and biological samples. Most importantly, the authors discover that CSRS has a much more favourable phase matching condition, with respect to CARS, for the backward scattered (epi) radiation. This is very important since epi detection is used in many emerging applications of coherent Raman microscopy, such as endoscopic configurations. This paper reports original findings which are important for the coherent Raman imaging community, especially for what concerns the epi detection modality. In my opinion, it deserves to be published in Nature Communications, after the authors have addressed the following points:

- 1) CARS microscopy is plagued by the non-resonant background (NRB), due to the electronic $\chi^{(3)}$ of the medium; is there a problem of NRB also in CSRS?
- 2) can the authors estimate the relative strength of the CSRS signal with respect to CARS, for similar excitation conditions?
- 3) in their current implementation of CSRS, the authors use blue/green excitation wavelengths. Can they comment on the sample damage that would occur in these conditions due to multi-photon absorption of the pump/Stokes beam?

Reviewer #3 (Remarks to the Author):

This manuscript describes an implementation of CSRS microscopy. The authors suppress the

strong fluorescence background with narrowband spectral filters and with doubly modulated lock-in detection. CSRS images are shown and the possibility of detecting strong epi-CSRS is discussed.

The manuscript is clearly written, the images of good quality and the Figures are carefully crafted. This is nice work, but the novelty and impact of the work do not meet the standards of Nature Communications. Several points of concern are given below.

- 1) CSRS may not have been explicitly shown before with the specific implementation of laser-scanning microscopy, but its broader implementation in microscopy was already shown in reference [11]. This earlier work makes the novelty of the current work less significant.
- 2) The use of double modulation is a well known general approach for isolating signals, and the use of narrowband spectral filters can not be classified as an original advance either.
- 3) The need for double modulation greatly complicates the CSRS technique relative to the much simpler CARS technique. Despite this added complexity, the images provided do not show a clear advantage over CARS type images. The current demonstration is insufficient in bringing out a practical advance over existing approaches in coherent Raman scattering microscopy.
- 4) The discussion on epi-CSRS is nice, but the effect appears small and it is not clear how much of an added advantage it brings to practical imaging experiments. In addition, only simulations are shown, lowering the impact of the current work.

Response to the reviewer comments

We thank all the reviewers for their constructive comments, which were taken into account when preparing the revised version of the manuscript.

Reviewer #1 (Remarks to the Author):

Heuke et al. present an implementation of CSRS where the pump and Stokes driving fields are independently modulated at f_1 and f_2 , and the coherent Stokes signal field is detected through a spectral notch filter at f_1-f_2 . They also perform calculations of the angular dependence of the signal field under standard high numerical aperture focusing and with pupil-filtered illumination.

The description of the experiment is clear, and the modulation scheme is similar to some of their previous work. Figure 3 shows that their fluorescence filtering approach is effective, and their signal field calculations suggest an interesting effect in an enhanced epi-emitted signal.

We thank this reviewer for his/her positive appreciation of our work.

I have a few comments:

1) While this is a very nice example of CSRS, it is not the first time that CSRS imaging has been demonstrated. Bito et al showed in 2013 that simultaneous CARS and CSRS imaging was beneficial for interpreting spectroscopic signals. That work should be cited: Bito et al., "Three-pulse multiplex coherent anti-Stokes/Stokes Raman scattering (CARS/CSRS) microspectroscopy using a white-light laser source, *Chemical Physics*, Volume 419, 2013.

We thank this referee for pointing to us this reference on CSRS which has been now added to the paper as Ref [12].

We respectfully note that, contrary to our work, this reference deals with CSRS signal in a spectroscopic context (using a spectrometer) and that the small image shown in Fig. 11 (41x41 pixels, long integration time of 200ms/pixel) is not a CSRS image but the spatially resolved difference of CARS and CSRS spectra after applying the maximum entropy method (MEM). Furthermore, the fluorescence background is not removed.

2) The diagram for CSRS shown in Figure 1 is different than I have seen elsewhere (such as in their reference 11 and in the Bito paper cited above). If this is a non-standard representation, perhaps they could comment on that.

Indeed, there are different representations of CSRS wave mixing process in the literature and its representation with 'up and down arrows' is somehow misleading. The order of the first 2 arrows (Stokes and pump beam) is of little importance because both colors are required simultaneously to match a molecular vibration. We have changed the representation of the CSRS process and follow the "Stimulated Raman Scattering Microscopy" textbook page 11 (<https://doi.org/10.1016/C2020-0-01880-3>), added now as Ref [14].

The new Figure 1 appears as:

Fig. 1. Coherent Raman imaging techniques in energy diagrams [14], relative radiation wavelength and energy conservation under plane-wave illumination.

3) The authors should indicate the time required to acquire the images. This information could be placed in the caption of Figure 2.

This has been corrected. We added to the caption of Fig. 3 "Pixel dwell time: 40 μ s. Image acquisition time: 40s (1000x1000)."

4) While the calculation results shown in Fig. 4 are interesting, it would be appropriate for the authors to demonstrate the effect they predict since it is the most unique aspect of the work.

Within our publication we demonstrate laser scanning CSRS imaging for the first time using a visible excitation. This is of high interest for the generation of coherent Raman images at high spatial resolution, because the CARS radiation is blue shifted and located within the UV while SRS is plagued by artifacts due all kinds of absorption effects.

The demonstration of the backscattering is currently impossible in this frequency range and necessitates to implement a CSRS experiment in the infrared range where the backward CSRS phase matching is possible. It is our plan to address this demonstration in a future work as it necessitates significant investments and experimental modification: we need an NIR excitation so that the incident angles of commercially available objective lenses are sufficient to direct the CSRS photons in the backward direction. The NIR excitation generates CSRS photons at 1500nm which cannot be detected with our current detectors. Furthermore, proving CSRS backscattering requires a descanned confocal detection. Thus, making our microscope fit for a concocal detection of backscattered CSRS photons at 1500nm requires significant changes that are beyond the scope of this paper.

Reviewer #2 (Remarks to the Author):

This paper provides the first experimental demonstration of a novel coherent Raman imaging modality, Coherent Stokes Raman Scattering (CSRS). In CSRS a light red-shifted with respect to the Stokes frequency ω_S is generated, according to the process $\omega_{CSRS} = 2\omega_S - \omega_{pu}$. Detection of the CSRS signal is challenged by its overlap with the sample fluorescence, generated by the pump/Stokes beams. Here the authors suppress the fluorescence by a combination of narrow bandpass filtering and modulation followed by lock-in detection, and demonstrate CSRS imaging on a series of test and biological samples. Most importantly, the authors discover that CSRS has a much more favourable phase matching condition, with respect to CARS, for the backward scattered (epi) radiation. This is very important since epi detection is used in many emerging applications of coherent Raman microscopy, such as endoscopic configurations.

This paper reports original findings which are important for the coherent Raman imaging community, especially for what concerns the epi detection modality. In my opinion, it deserves to be published in Nature Communications, after the authors have addressed the following points:

We thank this reviewer for his/her positive appreciation of our work.

1) CARS microscopy is plagued by the non-resonant background (NRB), due to the electronic $\chi^{(3)}$ of the medium; is there a problem of NRB also in CSRS?

Reviewer 2 is right, CSRS has a non-resonant background (NRB) just as CARS. The CSRS process $2\omega_s - \omega_p$ can also generate ω_{CSRS} without going through a vibrational state.

We added on page 9 (Methods: Experimental setup): "Note that the demodulation at f_1 - f_2 only removes the fluorescence background while the CSRS non-resonant four-wave-mixing background [31], that is inherent to all coherent Raman techniques, is still present. Nevertheless, the removal of this non-resonant background could be achieved by a heterodyne interference of the CSRS signal with a reference beam at the same wavelength [32] or by Kramers-Kronig or Maximum entropy based algorithms in application to CSRS spectra [13]."

2) can the authors estimate the relative strength of the CSRS signal with respect to CARS, for similar excitation conditions?

As outlined on page 17 in the book "Coherent Raman Scattering Microscopy" (CRC Press, 2013, <https://doi.org/10.1201/b12907>), the non-linear susceptibility of CARS, SRS and CSRS are identical in the absence of electronic resonances. Nevertheless, the intensity of the scattered radiation scales with ω to the power of four ($I_{as} \sim \omega_{as}^4$; $I_{cs} \sim \omega_{cs}^4$). For the excitation wavelength 449nm and 515nm matching the C-H stretching vibration at 2850 cm^{-1} , CARS and CSRS are radiated at 398nm and 604nm, respectively. The ratio of CARS and CSRS radiation is therefore $(604\text{nm} / 398\text{nm})^4 \approx 5$ meaning that the CARS radiation is 5 times stronger than the CSRS radiation. However, this theoretical CARS enhancement is lower in the experiment as standard detectors (e.g. PMT1001 Thorlabs) are 3 times more sensitive at 604nm (CSRS) than at 398nm (CARS). Going to shorter excitation wavelengths would be further advantageous for CSRS as the UV CARS radiation would be poorly transmitted through the collection optics, poorly detected by PMTs, and highly scattered by the sample.

We added on page 9 (Methods: Experimental setup): "We measured the CSRS and CARS (at 398nm) radiation strength for olive oil one after another (data not shown) and found comparable signal level. To avoid detector saturation for the acquisition of CSRS images, we applied the lowest possible PMT gain corresponding to an amplification of only $5 \cdot 10^3$."

3) in their current implementation of CSRS, the authors use blue/green excitation wavelengths. Can they comment on the sample damage that would occur in these conditions due to multi-photon absorption of the pump/Stokes beam?

Compared to the NIR excitation, we can use $2^4=16$ (ω_{cs}^4) times less power at the sample to receive the same signal level for the same focal spot size. Thus, our power levels at the sample range around 5mW to saturate our PMT (Thorlabs, PMT1001) at the lowest PMT gain possible (Gain: $5 \cdot 10^3$). These low power levels are also common in standard linear fluorescence microscopy and do not overheat the sample via linear absorption (in the absence of strong chromophores). All samples presented in Fig. 3 were scanned multiple times. We observed that the fluorescence background was reduced with scan time but the CSRS image demodulated at f_1 - f_2 remained unaffected from the first frame until frame number >100 indicating a negligible level of sample destruction due to multi-photon absorption.

To find the limit of VIS CSRS imaging, we investigated plant leaves. In this case, we observed for the first time burning of the sample which we attributed to the linear absorption of the pump beam at 449nm as a result of the absorption peak of chlorophyll b around 450nm.

We added to the manuscript page 9 (Methods: Experimental setup): "All samples were investigated in live image acquisition mode. Thus, some areas were scanned more than 100 times. We noticed that the fluorescence background signal within the DC channel was reduced over time as a result of photo-bleaching though the f1-f2 channel remained unaffected which indicates that our experimental conditions are below the damage threshold of ex vivo samples."

Reviewer #3 (Remarks to the Author):

This manuscript describes an implementation of CSRS microscopy. The authors suppress the strong fluorescence background with narrowband spectral filters and with doubly modulated lock-in detection. CSRS images are shown and the possibility of detecting strong epi-CSRS is discussed. The manuscript is clearly written, the images of good quality and the Figures are carefully crafted. This is nice work, but the novelty and impact of the work do not meet the standards of Nature Communications. Several points of concern are given below.

We thank this referee for his/her positive evaluation of the quality of our work. However, we respectfully disagree with his/her concern on the novelty and impact of our work and we provide below some arguments that, we believe, support our view.

1) CSRS may not have been explicitly shown before with the specific implementation of laser-scanning microscopy, but its broader implementation in microscopy was already shown in reference [11]. This earlier work makes the novelty of the current work less significant.

In reference [11] (<https://doi.org/10.1364/OL.34.000773>), Cui et al. shows NIR 3-color time-resolved pump probe spectroscopy to compare the signal strength of spontaneous Raman scattering with time-resolved CSRS. They use a time delayed scheme and a spectrometer to generate CSRS and spontaneous images of polystyrene beads. This complex experimental scheme (i) requires very long integration time (40ms/pixel), (ii) does not remove the fluorescence background and (iii) is not suitable to image biological samples - as demonstrated by the poor image quality of simple polystyrene beads. On the contrary our frequency modulated scheme, (i) uses a fast single pixel detector, (ii) removes the fluorescence background and (iii) provides high quality images of biological samples with a pixel dwell time of 40µs only. These advances render CSRS a viable imaging technique, for the first time.

2) The use of double modulation is a well known general approach for isolating signals, and the use of narrowband spectral filters can not be classified as an original advance either.

After a careful investigation, we could not find any publication that uses a double-modulation to reduce the fluorescence background in coherent Raman imaging. It is the combination of this modulation scheme for fluorescence rejection together with a narrow band detection and a single pixel detector that enables fast CSRS imaging.

3) The need for double modulation greatly complicates the CSRS technique relative to the much simpler CARS technique. Despite this added complexity, the images provided do not show a clear advantage over CARS type images. The current demonstration is insufficient in bringing out a practical advance over existing approaches in coherent Raman scattering microscopy.

The scope of this publication is to demonstrate that fast laser scanning CSRS is possible. We believe that our contribution is the necessary starting point to explore the CSRS advantages as compared to CARS such as its implementation with UV/VIS excitation wavelengths (as demonstrated here) but, most importantly, its unique ability to be phase-matched in the backward detection. This later point opens a completely new direction to perform chemical imaging in epi-direction.

4) *The discussion on epi-CSRS is nice, but the effect appears small and it is not clear how much of an added advantage it brings to practical imaging experiments. In addition, only simulations are shown, lowering the impact of the current work.*

Backscattering CSRS will be of high impact for various biomedical applications. It will facilitate the investigation of thick and opaque tissue samples. Furthermore, various applications will greatly benefit from an efficient backward radiation such as endoscopy, multi-focus coherent Raman imaging or laser scanning coherent Raman spectroscopy. We elaborate on this last example: Combining a spectrometer with a laser scanning coherent Raman microscope is currently impossible. CARS and SRS photons are forward directed and almost no light enters the entrance slit of a de-scanned spectrometer. Thus, current CARS and SRS spectroscopy are performed in forward direction while moving the sample instead of the laser beam which is a factor of >100 slower than using scanning galvo mirrors.

Backscattering CSRS should enable to combine fast spectrometer with microscopes to perform laser scanning imaging spectroscopy at speed levels of <math><10\mu\text{s}</math> per CSRS spectrum – see the figure to the right.

Backscattering CSRS laser scanning imaging spectroscopy will generate CSRS spectra at <math><10\mu\text{s}</math> per spectrum which is >100 times faster than state-of-the-art coherent Raman spectroscopy.

Reviewer #1 (Remarks to the Author):

The authors have met my concerns, and I feel that the manuscript is ready for publication.

Reviewer #2 (Remarks to the Author):

I have read the replies by the authors to my comments and those by the other reviewers and I think that they are satisfactory. I can therefore recommend this paper for publication in Nature Communications.

Reviewer #3 (Remarks to the Author):

The authors have revised the manuscript and have provided a response to the Reviewers' concerns. Although I believe this work is nicely carried out and may have some merit in the long run, the fact remains that the principle of CSRS microscopy has been shown before. The current implementation is a nice improvement on this approach rather than a conceptually new advance. The double modulation technique also complicates this particular approach relative to CARS, which is much easier to implement and provides the same information. The back-scattered CARS signal in tissues is strong and so there is no real practical benefit to the improved epi-CSRS signal (which is only shown theoretically here, not experimentally). In all, although I really like this manuscript, the impact and novelty do not rise to the level of a Nature Communications publication.

Response to the reviewer comments

We would like to thank again all the reviewers for their constructive comments and the overall positive evaluation.

Reviewer #1 (Remarks to the Author):

The authors have met my concerns, and I feel that the manuscript is ready for publication.

We thank reviewer 1 for his/her positive appreciation of our work.

Reviewer #2 (Remarks to the Author):

I have read the replies by the authors to my comments and those by the other reviewers and I think that they are satisfactory. I can therefore recommend this paper for publication in Nature Communications.

We thank reviewer 2 for his/her positive appreciation of our work.

Reviewer #3 (Remarks to the Author):

The authors have revised the manuscript and have provided a response to the Reviewers' concerns. Although I believe this work is nicely carried out and may have some merit in the long run, the fact remains that the principle of CSRS microscopy has been shown before. The current implementation is a nice improvement on this approach rather than a conceptually new advance. The double modulation technique also complicates this particular approach relative to CARS, which is much easier to implement and provides the same information. The back-scattered CARS signal in tissues is strong and so there is no real practical benefit to the improved epi-CSRS signal (which is only shown theoretically here, not experimentally). In all, although I really like this manuscript, the impact and novelty do not rise to the level of a Nature Communications publication.

We thank referee 3 for his/her positive evaluation of our work. However, we respectfully disagree again with his/her concern on the novelty and impact of our work. Backscattered epi-CARS is indeed strong but does not arise directly from the excitation laser focus, but is a volume scattering process. Epi-CSRS arises directly from the focus and can be refocused. This property will enable to place a spectrometer at the descanned position of a laser scanning microscope in backward direction to perform fast CSRS imaging spectroscopy. It will be also important for resolution enhancing CSRS image scanning microscopy as well as epi-detected multi-focus coherent Raman microscopy.